# Biochar Mitigated Yield-Scaled N₂O and NO Emissions and Ensured Vegetable Quality and Soil Fertility: A 3-Year Greenhouse Field Observation

Xi Zhang [1,2], Qianqian Zhang [1,3], Xintong Xu [1], Yubing Dong [1,4] and Zhengqin Xiong [1,*]

[1] Jiangsu Key Laboratory of Low Carbon Agriculture and GHGs Mitigation,
College of Resources and Environmental Sciences, Nanjing Agricultural University, Nanjing 210095, China;
2019203054@njau.edu.cn (X.Z.); qqzhang@zafu.edu.cn (Q.Z.); 2020203058@stu.njau.edu.cn (X.X.);
2021203058@stu.njau.edu.cn (Y.D.)

[2] Department of Soil Science of Temperate Ecosystems, Department of Agricultural Soil
Science, Georg-August University of Göttingen, Büsgenweg 2, 37077 Göttingen, Germany

[3] State Key Laboratory of Subtropical Silviculture, Zhejiang A&F University, Hangzhou 311300, China

[4] Huaiyin Institute of Agricultural Sciences of Xuhuai Region in Jiangsu,
Jiangsu Academy of Agricultural Sciences, Huaian 223001, China

* Correspondence: zqxiong@njau.edu.cn; Tel.: +86-25-8439-5148; Fax: +86-25-8439-5210

**Abstract:** Biochar amendments have been widely used in agricultural soil for lowering N₂O and NO emissions while improving soil fertility and crop performance. However, a thorough understanding of the longevity of the favorable effects would be a prerequisite for large-scale biochar application in agriculture. We conducted a three-year greenhouse vegetable trial in Southeast China to systematically investigate the impacts of biochar mixed with nitrogen (N) on soil N₂O and NO emissions, vegetable performance, and soil fertility at an interannual scale. The field experiment was established in November 2016 with biochar (0, 20 and 40 t ha$^{-1}$; C0, C1, and C2, respectively), applied once without/with 240 kg N ha$^{-1}$ urea (N0 or N1, respectively). Soil N₂O and NO emissions were monitored during the spring vegetable cultivation period, and vegetable yield, quality, and soil properties were measured after harvests in 2018, 2019, and 2020. Results indicated that N application significantly increased N₂O and NO emissions and vegetable yield throughout the trial period. Biochar combined with N generally reduced N₂O and NO emissions and emission factors while increasing the vegetable yield, leading to lower yield-scaled N₂O and NO emissions in 2018 and 2019. Biochar markedly enhanced soil pH and organic carbon and persisted, but generally had no significant effect on N use efficiency (NUE), vegetable quality, and soil fertility index (SFI) among treatments in over-fertilized vegetable fields. Based on our results, biochar application at 20 t ha$^{-1}$ combined with N seemed to achieve the highest agronomic and environmental benefits for intensive vegetable production in Southeast China.

**Keywords:** intensive vegetable production; biochar; nitrous oxide; nitric oxide; crop performance; soil fertility

## 1. Introduction

In China, greenhouse vegetable production (GVP) has been rapidly developing since the 1980s due to the growing population and vegetable demand [1]. The fields for GVP in China totaled 4.67 million hectares in 2018, accounting for about 83% of the world's GVP areas [2]. The GVP systems are characterized by high input of nitrogen (N) fertilizers and irrigation and frequent cropping indices [3,4]. Generally, annual N fertilizer application rates in GVP were 2–5 times greater than those in open fields, far exceeding the nutrient requirements for vegetables [3]. Thus, the negative environmental impacts triggered by the intensification of GVP have gradually become prominent [1]. Previous studies indicated that excessive N input disrupted nutritional balance, and lowered soil pH, microbial activity,

and diversity [5–7]. Furthermore, N fertilizer-induced soil nitrous oxide ($N_2O$, a long-lived and potent greenhouse gas) and nitric oxide (NO, atmospheric photochemical pollutant) emissions have received widespread attention owing to the robust and steady rise of emission factors [8]. Therefore, the integrated improvement of $N_2O$ and NO mitigation while enhancing crop production and soil fertility should be implemented to promote sustainable agricultural development.

Biochar, a carbon (C)-rich inert material pyrolyzed by biomass under oxygen-depleted conditions, has been proposed as the soil amendment for enhancing C sequestration, mitigating $N_2O$ and NO emissions, and improving soil fertility and plant growth [9–11]. Characteristics that make biochar an appropriate soil conditioner include liming effect, rich porosity, high C content and stability, large surface area and strong adsorption, and nutrient retention capacity [12]. Numerous mechanisms have documented that biochar amendment can positively mitigate $N_2O$ and NO emissions [13–15]. Recent meta-analyses revealed that biochar application decreased agricultural soil $N_2O$ and NO production by 38% and 8.3%, respectively [13,16]. Additionally, biochar interacting with N can enhance soil quality and crop production via (1) slowing soil acidification [17], (2) regulating soil structure and aggregate formation [18], (3) enhancing soil hydraulic properties [19], (4) improving nutrient recycling and utilization efficiency [20], (5) stimulating root growth and photosynthetic performance [21], or via a combination of the above mechanisms. However, due to various soil types, biochar properties, and experimental conditions, there is still no consensus on $N_2O$ and NO mitigation, crop production, and soil quality. For instance, Jeffery et al. [22] found that yield-stimulating effects of biochar are universal in weathered tropical soils compared with temperate soils. Contrastingly, some studies reported the positive impacts of biochar application in temperate soils with high fertility [4,23]. Biochar amendment, on the other hand, had exhibited neutral or fewer benefits upon incorporation in temperate soils [24,25]. Collectively, the effects of biochar on $N_2O$ and NO emissions, crop production, and soil quality require further studies.

Recently, Xiang et al. and Zhang et al. [10,26] reviewed the negative challenges of biochar on crop production, soil properties and biota, and associated environmental risk. The detrimental aspects are as follows: (1) release of toxic compounds and heavy metals in biochar [27], (2) reduction in the bioavailability of soil-applied agrochemicals owing to sorption behavior [28], (3) N immobilization due to a high C/N ratio, thereby limiting plant N uptake [29], (4) increase in native soil organic C priming [30], (5) shifts in native soil biota [31]. Moreover, the time impact of soil–biochar interactions on $N_2O$ and NO emissions, crop performance, and soil properties cannot be ignored [32,33]. Biochar after its soil incorporation has been confirmed to undergo different reactions, including the increase in O-rich functional groups and labile C substrate degradation [14,32]. Some studies have demonstrated that fresh and aged biochar amendment showed inconsistent results for $N_2O$ and NO emissions, crop growth, and soil biogeochemical cycling [12,34,35]. So far, however, mounting studies are mainly from short-term fields or incubation, and the interannual variations of biochar on $N_2O$ and NO emissions, crop performance, and soil quality after several years of a single biochar application remain largely unclear.

Admittedly, we cannot fully understand the impacts of biochar on crop growth and soil quality by knowing the status only once after biochar application. In this study, continuous measurements of its residual effect on $N_2O$ and NO production, vegetable yield, quality, and soil properties were carried out in a greenhouse vegetable field for three years. Meanwhile, the vegetable quality index and the soil fertility index (SFI), integrated by the specific vegetable quality and soil properties, were adopted to evaluate the vegetable quality and soil fertility. Thus, the objectives of this research were to quantify the interannual impacts of N interacted with biochar on $N_2O$ and NO emissions, vegetable performance, and soil properties in an intensive GVP system. We hypothesized that biochar application could make a positive impact on $N_2O$ and NO emission mitigation, crop performance, and SFI, and persist as time goes by. The results of our study could provide valuable guidance for relevant biochar application and sustainable development for GVP systems.

## 2. Materials and Methods

### 2.1. Site Description and Biochar Properties

The experiment was conducted at the Doucun site (32°01′ N, 118°52′ E) in Nanjing, Jiangsu Province, eastern China. This experimental site, established in November 2016, is a typical vegetable field that had been cultivated for almost twelve years. The local climate characteristics can be found in Zhang et al. [36]. The soil is classified as a *Haplic Luvisols*, and the soil properties before the biochar addition from the top 20 cm are shown in Table S1. Wheat straw-derived biochar applied in our study was obtained from Henan Sanli New Energy Co., Ltd. (Shangqiu, China). The production details and physicochemical properties of biochar are listed in Zhang et al. [36].

### 2.2. Field Design

Six treatments with three replicated plots (2 m × 2 m) were laid out in a completely randomized design: N0C0, N0C1 (20 t ha$^{-1}$ biochar), N0C2 (40 t ha$^{-1}$ biochar), N1C0 (urea), N1C1 (urea + 20 t ha$^{-1}$ biochar), N1C2 (urea + 40 t ha$^{-1}$ biochar). The biochar was applied once to the vegetable fields on 17 November 2016 and was mixed into the soil plow layer (0–20 cm). Before transplanting, urea (46.0% N), calcium magnesium phosphate (14.0% $P_2O_5$), and potassium chloride (63.2% $K_2O$) were applied at a rate of 240 kg ha$^{-1}$ crop$^{-1}$, respectively. Details of the applied chemical fertilizers are presented in Table S2. All management procedures (e.g., sowing, fertilizer, irrigation, harvest, and fallow) followed the local agronomic practices. Further information is available according to Zhang et al. [36].

### 2.3. Gases Sampling and Measurements

From 2018 to 2020, one vegetable crop was investigated in the experimental field each year to quantify the interannual variations of long-term biochar application. The baby bok choy (*Brassica rapa Chinensis* L.) was sown on 18 April 2018, 26 March 2019, and 24 May 2020, and harvested on 20 May 2018, 28 April 2019, and 17 June 2020. Soil $N_2O$ and NO were collected with the static chamber. Details concerning the gas collection and analyses can be found in our previous publication [37]. Briefly, gas samples were collected simultaneously with three replicates for each treatment at 9:00–11:00 a.m. every two or three days after fertilizer application. After sealing the chamber at 0, 10, 20, and 30 min, 20 mL of gas was taken from the chamber using a sealed syringe for each sealing time for $N_2O$ fluxes measurement. $N_2O$ concentrations were determined with a gas chromatograph (Agilent 7890A, Agilent Ltd., Shanghai, China), and fluxes were derived from the linear increases in gas concentration over time. Gas samples for NO flux measurement were removed from the same closed chamber at 0 and 30 min after chamber closure with a 1.0 L sampling bag (Delin Gas Packing Co., Ltd., Dalian, China). NO concentrations were analyzed by a model 42i chemiluminescence NO-$NO_2$-$NO_X$ analyzer (Thermo Environmental Instruments, Inc., Franklin, MA, USA), and fluxes were calculated by the concentration differences between the two collected samples. The cumulative gas emissions were obtained by calculating emissions averaged between every two adjacent intervals of gas collection. Additionally, soil temperature and moisture were measured according to Zhou et al. [37].

### 2.4. Vegetable Yield, NUE, Emission Factors, Yield-Scaled $N_2O$ or NO Emission, and Quality Analysis

Vegetable yields (fresh weight) in each plot were obtained by weighing all the above-ground vegetable parts. The vegetable samples were dried to obtain the dry matter yield at 65 °C. Plant N contents were tested by the Kjeldahl method. Then, N uptake was obtained from the sum of the N masses harvested in the biomass. The N use efficiency (NUE) was calculated by dividing the N uptake by the N application rate and the $N_2O$ or NO emission factor was calculated by dividing the cumulative $N_2O$ or NO emissions by applied fertilizer N during each harvest period for three years [4]. Additionally, the yield-scaled $N_2O$/NO

emissions were calculated as follows: yield-scaled $N_2O$ (or NO) emissions (g N $t^{-1}$ yield) = cumulative $N_2O$ (or NO) emissions/vegetable yield [4].

Vegetable quality, including nitrate, vitamin C, soluble sugar, and soluble protein analyses, were conducted in triplicate with fresh plant samples within one week [38]. The contents of nitrate and soluble sugar were tested using the salicylic acid and anthrone colorimetric technique methods after the baby bok choy was extracted in boiling water. The contents of vitamin C were analyzed using 2,6-dichlorophenol titration methods, and soluble protein contents were measured by the colorimetric method staining with Coomassie Brilliant Blue G-250.

Referring to Ke et al. [38], the comprehensive evaluation of vegetable quality was calculated as follows: $Q_i = \sum(X_i/X_{max}) \times 0.5 - \sum(Y_i/Y_{max}) \times 0.5$. Where $Q_i$ represents the comprehensive evaluation value of vegetable quality; $X_i$ and $Y_i$ are the nutritional quality items (including vitamin C, soluble sugar, and soluble protein) and sanitary quality item (nitrate), and $X_{max}$ and $Y_{max}$ are the maximum values of each item; 0.5 is the weight of each index.

### 2.5. Soil Collection and Analysis

After the baby bok choy harvest, soil samples were taken immediately from the surface layer (0–20 cm) on 20 May 2018, 28 April 2019, and 17 June 2020. For each plot, six soil cores were collected following the "S" pattern and sieved (2 mm) to obtain a composite sample. Every sample was separated into two parts after removing stones and plant and animal residues. One part was air-dried for analyzing conventional soil physicochemical parameters. The other part was kept at 4 °C for measuring soil inorganic N content, soil dissolved organic C/N (DOC/DON), and microbial biomass C/N (MBC/MBN). Details about the specific measurements are provided in the Supplementary Materials.

### 2.6. Overall Soil Fertility Assessment

The soil fertility index was obtained by integrating efficient soil fertility indicators based on the correlation ($p < 0.05$) with vegetable yield and the minimum data set (MDS) method was selected to calculate the SFI. Briefly, the first step was to select soil indicators for an MDS and then reduce the dimensions of the selected indicators according to the results of principal component analysis (PCA). Only the principal components (PCs) having eigenvalues $\geq 1$ and those that explained more than 5% of the variation in the data were selected for the MDS. The highly weighted (HW) factor loadings with absolute values within 10% of the highest factor loading were retained in the MDS for each PC reserved. When multi-variables existed in a PC and were highly correlated ($r > 0.70$), the variable was chosen for the MDS based on the highest norm value. Secondly, each variable chosen for MDS was transformed and normalized to a 0–1 scale using the standard scoring functions method. Multiple regression analysis and PCA were used to determine the weight of each MDS indicator, which was equal to the ratio of the standardized regression coefficient to the sum of all MDS indicator standardized regression coefficients, as well as the ratio of the community to the sum of all MDS indicator communities, respectively. Thirdly, the SFI was calculated according to the weighting factor and indicator score for each variable. According to Zhang et al. [39], details about the SFI methods and calculations are presented.

### 2.7. Statistics

Data statistical analyses and graphing were operated by SPSS v22.0 (IBM Co., Armonk, NY, USA) and OriginPro 2018 (OriginLab, Northampton, MA, USA). One-way analyses of variance (ANOVA) were used to assess the statistically significant difference in $N_2O$ and NO emissions, crop performance, and soil properties among different treatments based on Tukey's HSD test ($p < 0.05$). Two-way ANOVA and repeated-measures ANOVA were used to analyze the $N_2O$ and NO emissions, crop performance, and soil properties for the two factors (N and biochar) and their interactions ($p < 0.05$). The Pearson correlation was analyzed to identify relationships between measured soil indicators and vegetable yield.

## 3. Results

### 3.1. $N_2O$ and NO Fluxes and Cumulative Emissions

Two distinct dynamic patterns, depending on whether there was an N addition or not, were presented in the $N_2O$ and NO fluxes among treatments (Figure 1). The $N_2O$ and NO peak values appeared one week after fertilization and irrigation events for each crop period and then rapidly decreased to the background value. Two-factor repeated-measures ANOVA showed that time, N, and biochar had significant effects on cumulative $N_2O$ and NO emissions (Table S3, $p < 0.01$). N application greatly stimulated the $N_2O$ and NO emissions at each biochar level during the observation period (Figure 2a,b). Biochar application had no significant influence on the treatments without N. Compared with the N1C0, biochar interacted with N markedly reduced the $N_2O$ emissions by 23.6–40.0% and 17.7–36.4% and NO emissions by 16.7–25.1% and 18.2–34.8% in 2018 and 2019, respectively. However, no significant differences in $N_2O$ and NO emissions occurred among N1C0, N1C1, and N1C2 treatments in 2020.

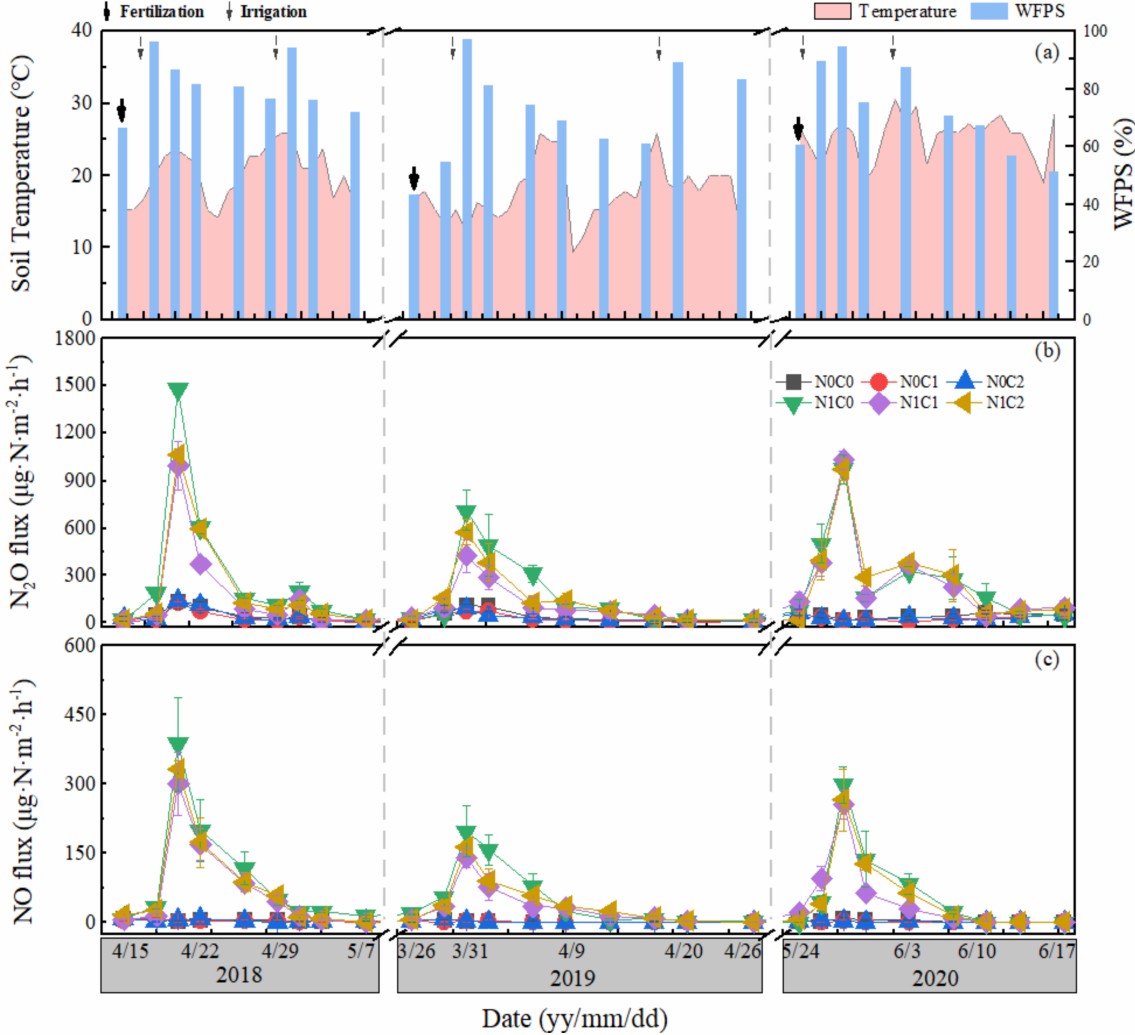

**Figure 1.** Dynamics of soil temperature and WFPS (**a**) and soil $N_2O$ ((**b**), *n* = 3) and NO ((**c**), *n* = 3) fluxes of each vegetable season across all treatments (N0C0, N0C1, N0C2, N1C0, N1C1, N1C2) from 2018 to 2020. The solid and dashed arrows denote fertilization and irrigation, respectively. Dashed vertical lines separate different vegetable crops. N0 or N1 represents N fertilization at 0 or 240 kg N ha$^{-1}$, respectively. C0, C1, and C2 represent biochar application at 0, 20, and 40 t ha$^{-1}$, respectively.

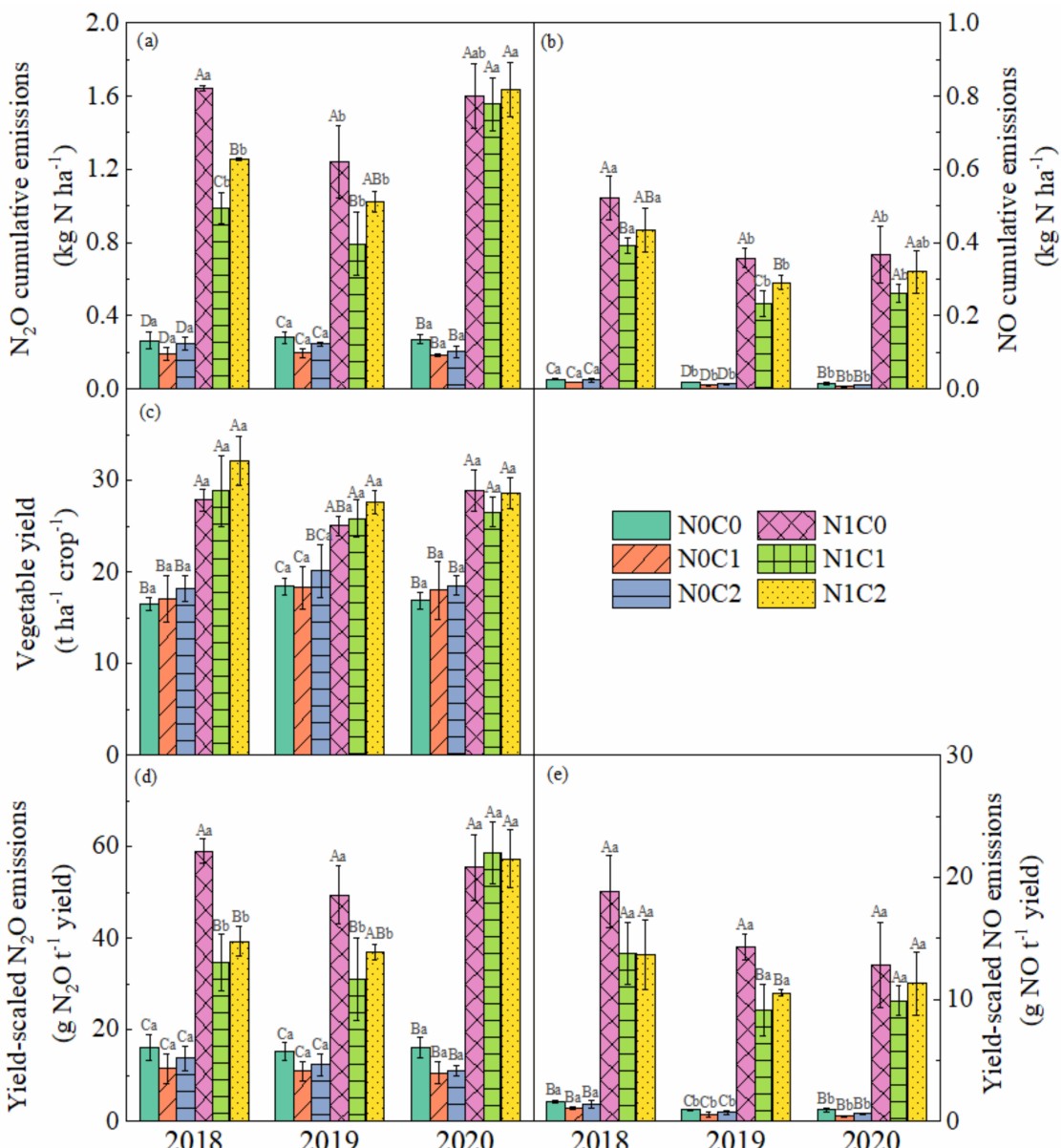

**Figure 2.** Cumulative $N_2O$ and NO emissions (**a,b**), vegetable yield (**c**), and yield-scaled $N_2O$ and NO emissions (**d,e**) of each vegetable season across all treatments from 2018 to 2020. Bar indicates the standard deviation (*n* = 3) of three replicates. Capital letters indicate significant differences among treatments for each cropping year at *p* < 0.05. Lowercase letters indicate significant differences among cropping years for each treatment at *p* < 0.05.

### 3.2. Vegetable Yield and Yield-Scaled Emissions

Total vegetable yields were different among treatments across three observed crop seasons, depending on whether N addition or not (Figure 2c, *p* < 0.05). N application significantly increased the vegetable yield at each biochar level by 68.7–76.8%, 35.9–41.4%, and 47.8–71.4% for three observed crop seasons. Biochar addition had no significant effect on vegetable yield at the same N level each year. However, compared with N1C0, the vegetable yield was higher by 3.60% and 15.3% and 3.25% and 10.5% for N1C1 and N1C2 in 2018 and 2019, respectively.

Integrating the emissions and vegetable yield results for the observation period showed that yield-scaled $N_2O$ and NO emissions ranged from 10.6–59.0 g N t$^{-1}$ and 0.40–18.8 g N t$^{-1}$ yield (Figure 2d,e). Compared with the treatments without N, significant increases in yield-scaled $N_2O$ emissions induced by N application were detected by

184–456% with the same biochar rates. On the contrary, biochar amendment significantly decreased yield-scaled $N_2O$ emissions by 33.5–41.1% and 25.2–37.4% in the N-fertilized treatments in 2018 and 2019, respectively, but had no significant influence in 2020, and the treatments without N, indicating a mitigation effect of biochar amendment when combined with the N application. For yield-scaled NO emissions, obvious increases induced by the N application were detected with the same biochar rates. The biochar amendment significantly lowered yield-scaled NO emissions in the N-fertilized treatments for all three observed crop seasons, although some of the differences were statistically non-significant.

### 3.3. $N_2O$ and NO Emission Factors and NUE

Direct $N_2O$-N emission factors ranged from 0.21 to 0.57% across the three observed crop seasons while being from 0.09 to 0.21% for NO-N emission factors (Figure 3a,b). Compared with the N1C0 treatment, biochar application markedly lowered the $N_2O$-N emission factor by 28.2–47.5% and 22.8–46.9% and NO-N emission factor by 17.6–26.5% and 19.1–36.6% in 2018 and 2019, respectively. However, biochar did not affect $N_2O$-N and NO-N emission factors in 2020 (relative to the N1C0).

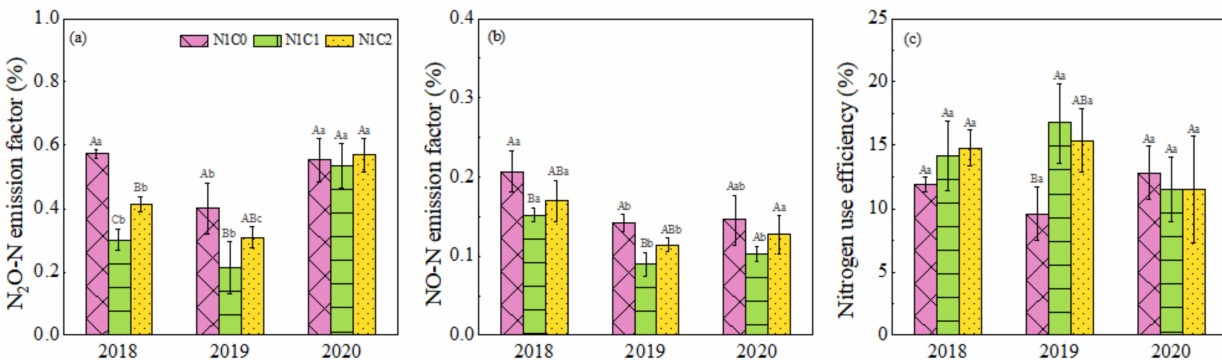

**Figure 3.** $N_2O$ and NO emission factors (**a**,**b**) and nitrogen use efficiency (**c**) of each vegetable season across all treatments from 2018 to 2020. Bar indicates the standard deviation ($n = 3$) of three replicates. Capital letters indicate significant differences among treatments for each cropping year at $p < 0.05$. Lowercase letters indicate significant differences among cropping years for each treatment at $p < 0.05$.

The NUE among treatments was low in this study, ranging from 9.60 to 16.8% (Figure 3c). The N1C1 and N1C2 treatments increased NUE by 19.1–24.2% and 60.3–74.6% in 2018 and 2019, respectively, but were non-statistically significant in 2020 despite a 10.2–10.3% decrease ($p > 0.05$).

### 3.4. Vegetable Quality

The vegetable nitrate content was different among treatments across three observed crop seasons (Figure 4a, $p < 0.05$). N application increased the nitrate content at each biochar level by 36.9–88.3%, 38.3–97.9%, and 53.3–104% for each crop season. However, the biochar addition had no obvious effect on vegetable nitrate content among N1C0, N1C1, and N1C2 treatments. No significant differences in vegetable nutritional qualities (vitamin C, soluble sugar, soluble protein) were found among the six treatments (Figure 4). Two-way repeated-measures ANOVA for vegetable yield and quality were presented in the Supplementary Materials (Table S4). For the vegetable quality index, no significant changes were found among treatments for each crop season (Figure 4e).

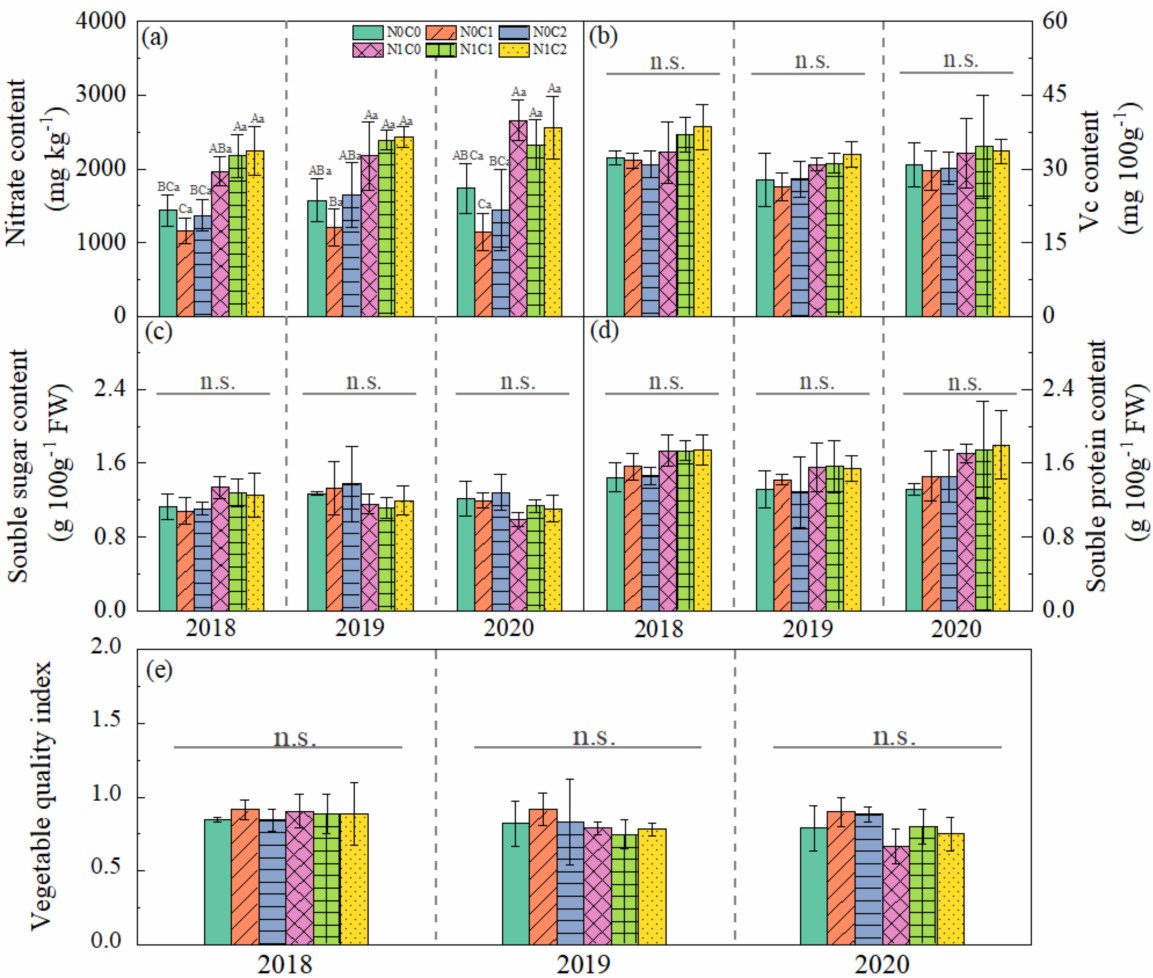

**Figure 4.** Vegetable quality ((**a**) nitrate, (**b**) vitamin C, (**c**) soluble sugar, (**d**) soluble protein, (**e**) quality index) of each vegetable season across all treatments from 2018 to 2020. The bar indicates the standard deviation (*n* = 3) of three replicates. Capital letters indicate significant differences among treatments for each cropping year at *p* < 0.05. Lowercase letters indicate significant differences among cropping years for each treatment at *p* < 0.05. n.s. means no significant difference across all treatments at 0.05 levels.

### 3.5. Soil Properties and SFI

The results of two-factor repeated-measures ANOVA indicated that time showed a significant impact on soil properties, except for pH (Table 1, *p* < 0.05). The pH, EC, TN, $NO_3^-$-N, available K, DOC, DON, and MBC were markedly influenced by N, biochar, and their interactions. N application increased TN, $NO_3^-$-N, DON, and MBN but decreased pH at the same biochar levels for each crop season (Table S5). Biochar application significantly increased SOC by 9.25–25.8%, 11.0–40.7%, and 6.34–14.4% but decreased CEC by 6.50–16.6%, 6.79–14.0%, and 4.91–15.9% in 2018, 2019, and 2020, respectively (Table S5, *p* < 0.05). Additionally, soil pH values improved with increasing the biochar application rate among N1C0, N1C1, and N1C2 treatments across the three observed crop seasons. Two-way ANOVA for soil properties in each crop season is listed in Supplementary Materials (Table S6).

**Table 1.** Results of repeated-measures ANOVA (F-values and significance) for soil physiochemical properties. *** Significant at $p < 0.001$; ** Significant at $p < 0.01$; * Significant at $p < 0.05$.

| Parameter | Year | N | Biochar | Year | N × Biochar | N × Year | Biochar × Year | N × Biochar × Year |
|---|---|---|---|---|---|---|---|---|
| CEC | 2018–2020 | 0.507 | 114.4 *** | 80.94 *** | 4.015 * | 0.638 | 1.133 | 1.701 |
| pH | 2018–2020 | 167.4 *** | 56.95 *** | 3.329 | 40.43 *** | 5.792 ** | 2.642 | 1.556 |
| EC | 2018–2020 | 2374 *** | 214.1 *** | 1270 *** | 95.80 *** | 301.2 *** | 35.05 *** | 7.512 *** |
| SOC | 2018–2020 | 0.335 | 114.1 *** | 5.127 * | 5.249 * | 0.126 | 9.106 *** | 2.371 |
| TN | 2018–2020 | 178.4 *** | 15.97 *** | 7.957 ** | 10.59 ** | 0.770 | 0.177 | 6.365 ** |
| TP | 2018–2020 | 6.406 * | 7.969 ** | 6.121 ** | 0.712 | 0.327 | 0.978 | 0.703 |
| $NH_4^+$-N | 2018–2020 | 3.234 | 0.585 | 735.8 *** | 1.136 | 2.345 | 2.728 | 6.129 ** |
| $NO_3^-$-N | 2018–2020 | 1439 *** | 56.36 *** | 609.8 *** | 72.99 *** | 422.1 *** | 77.73 *** | 62.45 *** |
| Available P | 2018–2020 | 4.701 | 15.15 ** | 192.3 *** | 2.622 | 12.04 *** | 7.431 *** | 4.396 ** |
| Available K | 2018–2020 | 86.97 *** | 75.67 *** | 16.12 *** | 178.3 *** | 31.49 *** | 20.73 *** | 18.12 *** |
| DOC | 2018–2020 | 41.11 *** | 4.225 * | 7.171 ** | 4.908 * | 16.31 *** | 9.590 *** | 2.808 * |
| DON | 2018–2020 | 1487 *** | 5.546 * | 620.1 *** | 6.340 * | 439.8 *** | 7.194 ** | 2.317 |
| MBC | 2018–2020 | 10.29 ** | 25.30 *** | 70.06 *** | 6.108 * | 0.512 | 3.475 * | 4.305 ** |
| MBN | 2018–2020 | 71.57 *** | 1.619 | 16.71 *** | 0.001 | 1.616 | 1.497 | 2.266 |

Note: CEC: cation exchange capacity; EC: electrical conductivity; SOC: soil organic carbon; TN: total N; TP: total P; DOC: dissolved soil organic carbon; DON: dissolved soil organic nitrogen; MBC: microbial biomass carbon; MBN: microbial biomass nitrogen.

As pH, EC, TN, TP, $NO_3^-$-N, available K, DOC, DON, and MBN were significantly correlated with vegetable yield (Table 2; $p < 0.05$); these soil parameters were chosen as efficient soil indicators for subsequent PCA analysis. The total variance of contributions from the first two PCs reached 63.6% (Table 3). The highly-weighted indicators were DON, $NO_3^-$-N, pH, and DOC within PC1. However, due to the strong correlations between DON and $NO_3^-$-N (Table S7, r > 0.70), only DON was selected for the MDS according to the norm value. The highly-weighted indicators were the available K and TP under PC2, similar to the MDS selection principle of PC1. In summary, DON, pH, DOC, available K, and TP were finally selected into the MDS (Table 3), which were then weighted and scored to obtain the SFI.

Generally, the two-factor repeated-measures ANOVA indicated that time and N have a significant impact on SFI (Table S4, $p < 0.05$). The N application significantly increased the SFI at each biochar level by 38.2–113% and 16.0–66.1% in 2018 and 2019, respectively (Figure 5). However, the biochar addition had no significant effect on SFI among treatments, except for 2020. Surprisingly, a significant decrease along the three crop seasons was observed among N1C0, N1C1, and N1C2 treatments.

**Table 2.** Correlation coefficients between soil properties and vegetable yield. ** Significant at $p < 0.01$; * Significant at $p < 0.05$.

| Soil Properties | Yield | Soil Property | Yield |
|---|---|---|---|
| CEC | −0.064 | $NO_3^-$-N | 0.655 ** |
| pH | −0.456 ** | Available P | −0.143 |
| EC | 0.457 ** | Available K | 0.279 * |
| SOC | 0.169 | DOC | −0.355 ** |
| TN | 0.725 ** | DON | 0.658 ** |
| TP | 0.341 * | MBC | −0.118 |
| $NH_4^+$-N | 0.088 | MBN | 0.634 ** |

Note: CEC: cation exchange capacity; EC: electrical conductivity; SOC: soil organic carbon; TN: total N; TP: total P; DOC: dissolved soil organic carbon; DON: dissolved soil organic nitrogen; MBC: microbial biomass carbon; MBN: microbial biomass nitrogen.

**Table 3.** Results of the principal component analysis of selected soil indicators and estimated communality and weight values of each soil property.

| Soil Properties | PC1 | PC2 | Norm | Communality |
|---|---|---|---|---|
| DON | 0.945 | −0.004 | 1.998 | 0.893 |
| $NO_3^-$-N | 0.908 | 0.138 | 1.926 | 0.844 |
| MBN | 0.786 | 0.231 | 1.681 | 0.670 |
| EC | 0.782 | 0.211 | 1.670 | 0.656 |
| TN | 0.750 | −0.073 | 1.589 | 0.568 |
| PH | −0.583 | 0.510 | 1.359 | 0.600 |
| DOC | −0.576 | 0.095 | 1.222 | 0.341 |
| Available K | 0.382 | −0.729 | 1.148 | 0.677 |
| TP | 0.379 | 0.575 | 1.028 | 0.474 |
| Eigenvalue | 4.470 | 1.253 | | |
| % of variance | 49.67 | 13.921 | | |
| Cumulative variance % | 49.67 | 63.593 | | |

Note: EC: electrical conductivity; TN: total N; TP: total P; DOC: dissolved soil organic carbon; MBN: microbial biomass nitrogen.

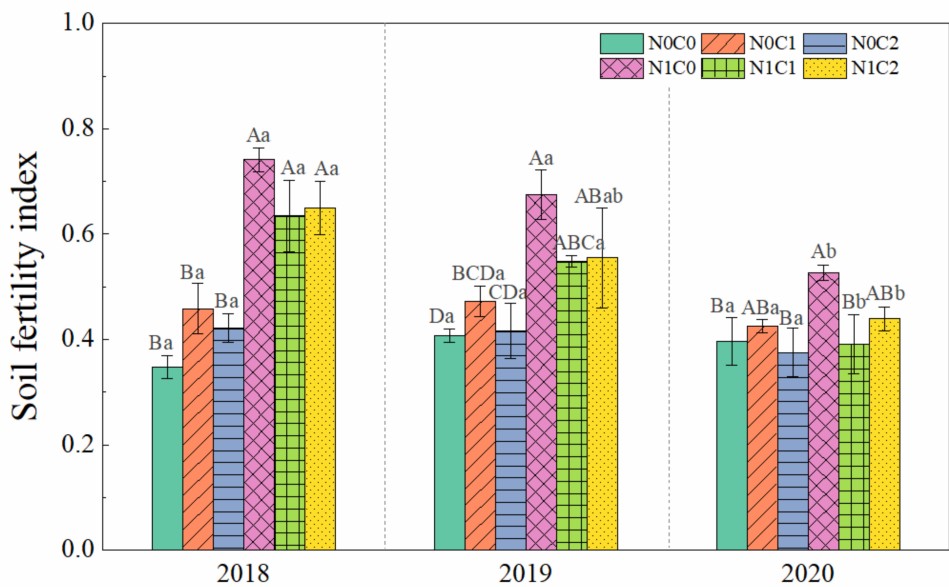

**Figure 5.** Soil fertility index of each vegetable season across all treatments from 2018 to 2020. The bar indicates the standard deviation (*n* = 3) of three replicates. Capital letters indicate significant differences among treatments for each cropping year at *p* < 0.05. Lowercase letters indicate significant differences among cropping years for each treatment at *p* < 0.05.

## 4. Discussion

### 4.1. Lasting Effect of Biochar on N₂O and NO Emissions

As generally accepted, biochar soil implementation could significantly lower $N_2O$ emission in agroecosystems [40,41], but the size effect for vegetable soils differed across the field studies [4,11,42–44]. During the vegetable growing season in 2018 and 2019, biochar interacted with N and significantly mitigated $N_2O$ emissions by 23.6–40.0% and 17.7–36.4%, respectively (Figure 2a). This finding was consistent with a meta-analysis, which showed biochar could reduce soil $N_2O$ emissions by 12–44% under field conditions [45]. Furthermore, our findings also confirmed that applying straw-derived biochar at 1% and 2% markedly reduced $N_2O$ emissions [41], while the mitigation effect of 40 t ha$^{-1}$ biochar was lower than that of 20 t ha$^{-1}$ biochar. Coincidently, Li et al. and Zhang et al. [11,42] also reported that 20 t ha$^{-1}$ biochar has a higher potential to reduce $N_2O$ emissions than 40 t ha$^{-1}$ biochar application.

N application and biochar significantly impacted $N_2O$ emissions, except in 2020, both separately and in combination, primarily by affecting $N_2O$ production processes and $N_2O$ product ratios [4,33]. During the growing season of vegetables, soil WFPS was mostly high (Figure 1a), which could stimulate nitrifiers and denitrifiers to generate $N_2O$ [46]. The reduction in biochar-induced $N_2O$ emissions was attributed to several mechanisms in our study. First, biochar lowered $N_2O$ production via immobilizing and adsorbing the available $NH_4^+$ for nitrification [47]. Additionally, previous research reported that biochar could decrease soil $NH_4^+$ by stimulating $NH_3$ emissions due to the liming effect [48]. Second, the release of toxic compounds of biochar into soils, inhibiting nitrifiers or denitrifiers, could be another critical regulator of $N_2O$ production [26]. Third, biochar-induced decreases in $N_2O$ emissions could be due to accelerated reduction in $N_2O$ into $N_2$ via promoting electron transfers to denitrifiers and $N_2O$ reductase gene expression [10].

However, the mitigation impact of biochar on $N_2O$ emissions disappeared in 2020 with biochar aging (Figure 2a), and similar results were also reported in other field experiments [24,35,49]. This finding was in line with a recent meta-analysis that biochar-induced decreases in $N_2O$ emissions were transient [13]. As mentioned by Cayuela et al., and Duan et al. [50,51], the changes in biochar over time (e.g., degradation, mineralization, and adsorption of organo-mineral layers) reduced the reactive organic functional groups on the biochar surface and the capacity to absorb fertilizer-derived $NH_4^+$ and $NO_3^-$ in biochar amended soil, leading to a lower biochar $N_2O$ mitigation effect. Meanwhile, the polycyclic aromatic hydrocarbons from biochars were absorbed by the surrounding soil after four years of field aging [52]. Although biochar is rich in labile C and N substances, only recalcitrant C remains in biochar-treated soil after 3.5 years of frequent irrigation and tillage, making it difficult for microbes to utilize [36].

For NO production, the fluxes and the cumulative emissions presented similar patterns to the $N_2O$ (Figures 1c and 2b). Biochar interacted with N significantly mitigated NO emissions by 16.7–25.1% and 18.2–34.8% in 2018 and 2019, respectively, while no reduction effect was observed in 2020 (Figure 2b). Nitrification and denitrification are the classically dominant pathways for soil NO production [53]. Generally, nitrification was considered the primary pathway of NO production in alkaline soil [14,15,54]. As mentioned above, biochar could promote the immobilization of $NH_4^+$, reducing N availability for NO generation by nitrifiers [47]. However, given the low molar $NO/N_2O$ emission ratios (<1), denitrification might be the primary process for soil $N_2O$ and NO emissions [55,56]. Overall, due to a lack of dynamic data on soil inorganic nitrogen and the abundance of microbial functional genes, the relationship between the mitigating impact of biochar on $N_2O$ and NO and biochar aging could not be effectively evaluated, and more studies are needed in the future.

High yearly variability of cumulative $N_2O$ and NO emissions existed across the three vegetable growing seasons. $N_2O$ and NO emissions in 2019 were low, with $N_2O$ and NO emission factors ranging from 0.21–0.40% and 0.08–0.14%, respectively (Figure 3). We speculated that low soil temperature restrained $N_2O$ and NO generation, as reported by Wu et al. [23] and Liao et al. [33], in turn masking the mitigating impact of biochar on $N_2O$ and NO production. Since few studies have attempted to explore the persistence of $N_2O$ and NO emissions suppressing effects in the GVP ecosystem [40], our understanding of whether a single biochar addition would be sufficient to mitigate soil $N_2O$ and NO production, in the long run, remains elusive.

*4.2. Lasting Effect of Biochar on Vegetable Yield, Yield-Scaled $N_2O$ and NO Emissions, Emission Factors, and NUE*

Yield increases have been documented as a direct advantage of biochar use in agriculture [40,57]. During a 3-year field trial, we monitored that N addition markedly stimulated vegetable yield at the same biochar addition rate, and biochar had no effect on vegetable yield at each N level. However, biochar had a substantial stimulatory effect on vegetable yield in 2018 and 2019, although the difference was insignificant (Figure 2c; Table S4). This finding seems contradictory to the reports of previous studies [4,11,42], which reported

that biochar increased vegetable yield by roughly 20–30% on average. However, Liao et al. and Mehmood et al. [33,58] reported that biochar application had no apparent stimulatory effect on corn or early rice yield during the observation period. This discrepancy may be attributed to the differences in soil properties, especially the alkaline characteristics of the tested soil, leading to a lower response to yield increase than in other soils. A recent meta-analysis also found that biochar had a higher effect on yield increases in acidic soils than in neutral or alkaline soils [22]. Another reason could be that the additional available nutrients from biochar have been degraded with aging [32]. Furthermore, any possible biochar-induced improvement in vegetable yield could have been masked, particularly because the N fertilizer application rate (240 kg N ha$^{-1}$ crop$^{-1}$, average of four crop seasons per year) was higher than the suggested optimum N magnitude in the vegetable fields (mean: ~762 kg N ha$^{-1}$ yr$^{-1}$) [2].

Analyzing $N_2O$ and NO emissions on a yield basis provides useful information for assessing the environmental impacts of intensive GVP systems. As shown in Figure 2d,e, yield-scaled $N_2O$ and NO emissions for baby bok choy were lower than the values for other vegetables previously reported [4,11,59]. This discrepancy may be attributed to low gas emissions caused by differences in the short growing period and low soil temperatures in spring. Furthermore, all aboveground portions of the leafy vegetable plants were considered as the yield, resulting in low values of yield-scaled $N_2O$ and NO in our vegetable field [59]. Overall, biochar significantly decreased yield-scaled $N_2O$ emissions in the N-fertilized treatments in 2018 and 2019, contributing to the $N_2O$-reducing and vegetable-increasing effects of biochar in an intensively managed vegetable field, while it had no significant influence in 2020 and the treatments without N. Our results indicated that the yield-scaled $N_2O$ emissions were minimal under the N1C1 treatment in the presence of N addition in 2018 and 2019 ($34.8 \pm 6.10$ vs. $31.0 \pm 9.08$ g $N_2O$-N t$^{-1}$ yield). The N1C1 treatment showed the lowest cumulative $N_2O$ emissions and the second-highest vegetable yield in 2018 and 2019. For yield-scaled NO emissions, biochar lowered yield-scaled NO emissions in the N-fertilized treatments for all three years, especially in the N1C1 treatment. Similarly, Li et al. [11] demonstrated that the 20 t ha$^{-1}$ biochar had the lowest yield-scaled $N_2O$ emissions from ultisols in an intensive vegetable field in South China.

In our study, $N_2O$-N and NO-N emission factors ranged from 0.21% to 0.57% and 0.09% to 0.21%, which was much lower than the latest results reported by Ma et al. [8]. We speculated that this may be due to the lower production of $N_2O$ and the higher reduction ratio, resulting in a net reduction in $N_2O$ and NO emissions. Additionally, it was found that biochar could generally decrease the $N_2O$ and NO emission factors, with the lowest value under the N1C1 treatment during the observation period. The biochar-induced NUE stimulation effect was minimal since biochar was ineffective at enhancing vegetable yield. Similarly, biochar amendment also increased, albeit not significantly, NUE in 2018 and 2019 compared with the N1C0 treatment (Figure 3c). Overall, the N1C1 treatment showed a great advantage in terms of the yield-scaled $N_2O$ and NO emissions, emission factors, and NUE.

### 4.3. Lasting Effect of Biochar on Vegetable Quality and Soil Fertility

Nitrate accumulation in vegetables is generally influenced by the amount and type of nutrients in the soil, as well as the amount, timing, and composition of fertilizers used. After being applied to farmlands, the majority of urea is converted to ammonia nitrogen quickly and then transformed into nitrate by biological nitrification in the soil. Thus, obvious nitrate accumulation in vegetables was observed due to the adsorption and storage of vegetables [60]. The N addition significantly increased the nitrate content at each biochar level, whereas nitrate content was unaffected by biochar during the 3-year field trial. Additionally, the contents of vitamin C, soluble sugar, soluble protein, and the vegetable quality index in all three seasons were hardly affected by biochar and N addition among treatments (Figure 4). Although Shi et al. [61] had confirmed that biochar may have a stimulating effect on the quality of vegetables, this phenomenon was not detected in our

study. However, Ke et al. [38] proved the effect of biochar addition on the quality index increase of pakchoi in a 40-d pot experiment. Thus, we speculated that biochar amendment may have exhibited only minor effects on vegetable quality due to the long-term exposure to soil. Despite the lack of a biochar-induced effect on vegetable quality in our study, a lesser understanding of how long-term biochar application affects the vegetable quality and the relevant mechanisms underlines that more research is required.

Numerous soil physicochemical properties changed with the aging of biochar, which indirectly affected the change in the soil fertility index (Table 1 and Table S4). Positive effects of biochar on soil properties (e.g., physical, chemical, and biological) have been widely reported in short-term studies [24,40,62]. The liming property of biochar caused appreciable changes in pH following biochar treatment for 3.5-years in our study, confirming that biochar application delayed the soil acidification process [63]. The SOC of biochar-amended soil was significantly higher, and the CEC was lower; however, there appeared to be no discernible pattern for the other indicators. Despite the fact that tillage and irrigation were frequent in GVP systems, the SOC enhancement owing to the direct input of recalcitrant C from biochar increased expectedly (Table S5). Wang et al. and Zhang et al. [9,36] also found that biochar with high aromatic compound fractions is resistant to biodegradation and significantly promotes soil C sequestration. Soil CEC showed a negative trend after biochar application, contrary to other research results [64]. Biochar underwent an aging process after tillage and irrigation, including the protonation and elimination processes of negative surface charges and a breakdown of the rich surface area structure [32]. Bakshi et al. [65] also demonstrated that soil CEC was reduced following maize stover biochar aging in field trials. Furthermore, N fertilizer addition increased TN, $NO_3^-$-N, DON, and MBN at the same biochar levels each year, but the advantages of biochar to enhance soil nutrients have not been well documented in this work. The corresponding mechanisms need to be further investigated.

The SFI was adopted in our work to assess trends in soil quality by integrating fourteen measured soil chemical parameters into a single index number. N application significantly increased the SFI at each biochar level in 2018 and 2019 (Figure 5). However, the biochar addition generally had no significant effect on SFI among treatments, while a significant interannual decrease was observed among N1C0, N1C1, and N1C2 treatments. This result revealed that continuous vegetable cultivation increased the risk of soil quality degradation. This finding seemed contradictory to the reports of previous studies [39,62,66,67], which reported that biochar generally improved overall soil quality. However, lower SFI values were observed in two contrasting soils [68], related to the soil's weathered condition and lower levels of measured indicators. Surprisingly, biochar soil amendment at different dosages did not produce distinct differences in SFI values across the three crop growing seasons. This phenomenon implied that biochar efficiency was intimately linked to soil fertility, which might be easily influenced by irrigation and fertilization. Additionally, the response of crop performance to soil fertility could be affected by soil physicochemical and biological properties [39]. Given the long-term overdose of nutrients to vegetable soil, it seemed that the impact of biochar on soil improvement was weak in our vegetable fields. Liu et al. and Medyńska-Juraszek et al. [24,25] discovered that biochar had limited benefits for rice yield and soil properties based on the field experiment (>3 years). Thereby, standardization or recommendation for biochar production conditions and application rates appropriate for soil fertility and crop productivity improvement will be necessary [63]. Undeniably, the value of SFI we estimated was subject to some uncertainties. On the one hand, we only considered the changes in chemical properties of soils after biochar addition, ignoring the variations in physical and biological properties. On the other hand, the short growth periods of vegetable crops with frequent irrigation and tillage might cause certain disturbances. Notably, we should combine numerous indicators to calculate the soil fertility index for assessing the soil quality more thoroughly in future studies.

## 5. Conclusions

This study revealed that biochar interacting with N markedly reduced $N_2O$ and NO emissions and stimulated the vegetable yield, resulting in the lower yield-scaled $N_2O$ and NO emissions in 2018 and 2019. Biochar generally decreased the $N_2O$ and NO emission factors, but the biochar-induced NUE stimulation effect was minimal during the observation period. Although biochar provided limited benefits for improving vegetable quality, soil nutrient content, and SFI in the fertile vegetable soils, delaying soil acidification and enhancing soil C sequestration were indisputable. Overall, 20 t $ha^{-1}$ of biochar amendment seems to be recommendable to foster sustainable development in the intensive GVP system in Southeast China, based on agronomic and environmental benefits. However, long-term studies conducted in situ will be necessary to fully evaluate the variations in soil, plant, and $N_2O$ and NO emissions, and potential risks for vegetable quality and the environment associated with various biochar applications.

**Supplementary Materials:** The following supporting information can be downloaded at: https://www.mdpi.com/article/10.3390/agronomy12071560/s1, [69,70] Table S1: Initial properties of the soil in our study (mean $\pm$ SD, n = 3); Table S2: Main compounds and ingredient content of the chemical fertilizers applied; Table S3: Results of two-way ANOVA and repeated-measures ANOVA (F-values and significance) for cumulative $N_2O$ and NO emissions. *** Significant at $p < 0.001$; ** Significant at $p < 0.01$; * Significant at $p < 0.05$; Table S4: Results of two-way ANOVA and repeated-measures ANOVA (F-values and significance) for vegetable yield and quality and soil fertility index (SFI). *** Significant at $p < 0.001$; ** Significant at $p < 0.01$; * Significant at $p < 0.05$; Table S5: Soil physicochemical properties after vegetable harvest in 2018, 2019 and 2020 (mean $\pm$ SD, n = 3); Table S6: Results of two-way ANOVA (F-values and significance) for soil properties. *** Significant at $p < 0.001$; ** Significant at $p < 0.01$; * Significant at $p < 0.05$; Table S7: Pearson correlation coefficients among selected soil indicators. ** Significant at $p < 0.01$; * Significant at $p < 0.05$.

**Author Contributions:** Conceptualization, X.Z. and Z.X.; methodology, Q.Z. and X.X.; software, X.Z.; validation, X.X. and Y.D.; formal analysis, X.X.; investigation, X.Z. and Y.D.; resources, Z.X.; data curation, X.Z. and Q.Z.; writing—original draft preparation, X.Z. and X.X.; writing—review and editing, X.Z. and Z.X.; visualization, X.Z. and Q.Z.; supervision, Z.X.; project administration, Z.X.; funding acquisition, Z.X. and Y.D. All authors have read and agreed to the published version of the manuscript.

**Funding:** This research was jointly funded by the National Natural Science Foundation of China (41977078, 32001213) and the Postgraduate Research & Practice Innovation Program of Jiangsu Province, China (KYCX20_0591).

**Data Availability Statement:** All data included in this study are available upon request by contact with the corresponding author.

**Acknowledgments:** We thank Yanfeng Song, Haojie Shen, Xueyang Jiang for their assistance with the experimental analysis. The first author also thanks the China Scholarship Council for providing funds for him to pursue his study in Germany.

**Conflicts of Interest:** The authors declare no conflict of interest.

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
