# Peer review of "Biochar Mitigated Yield-Scaled N2O and NO Emissions and Ensured Vegetable Quality and Soil Fertility: A 3-Year Greenhouse Field Observation"

_agronomy, doi:10.3390/agronomy12071560_

Round 1

Reviewer 1 Report

I have read this article very carefully. Its quality is very high.

The introduction of the article is very brief and clear. The reader will learn all the necessary information he needs to understand the topic. The methodology of the article is very well designed and elaborated. The authors chose appropriate statistical methods.

The discussion is clear, the results are well and factually discussed. The article concludes with a clear and comprehensible conclusion.

The article deals with a very current topic. The use of N fertilizers causes major environmental problems. Any effort to mitigate these negative effects is very valuable.

The only thing that can be criticized is that 3 years is a short time to show all the effects of biochar on soil and yield. For example, in the temperate zone, where we studied the effect of biochar on soil quality and grain production, the changes did not take effect until about 8 years later. However, this complaint does not reduce the quality of the derivative work.

Thank you for a very good article.

Author Response

(1) The introduction of the article is very brief and clear. The reader will learn all the necessary information he needs to understand the topic. The methodology of the article is very well designed and elaborated. The authors chose appropriate statistical methods.

A: Thank you very much for your great support! We will continue to improve our researches in the future with your encouragement!

(2) The discussion is clear, the results are well and factually discussed. The article concludes with a clear and comprehensible conclusion.

A: Thank you very much for your great support!

(3) The article deals with a very current topic. The use of N fertilizers causes major environmental problems. Any effort to mitigate these negative effects is very valuable.

A: Thank you very much for your recognition of our work and constructive comments!

(4) The only thing that can be criticized is that 3 years is a short time to show all the effects of biochar on soil and yield. For example, in the temperate zone, where we studied the effect of biochar on soil quality and grain production, the changes did not take effect until about 8 years later. However, this complaint does not reduce the quality of the derivative work.

A: Yes, you are completely right! Thanks for your insightful review.

As you mentioned, a 3-years field observation still belongs to short term effects to assess the impacts of biochar addition on soil properties and vegetable yields. Considering that the current publications are mainly from incubations or short-term field studies (< 2 years) (Schmidt et al., 2021; Zhang et al., 2019; Zhang et al., 2021). The intensified vegetable cultivation in China is quite different from other crop production systems (Li et al., 2017). Thus, our results are pointing out the necessity of medium- and long-term effects of biochar amendments in vegetable fields. We will continue to investigate and follow up on the residual effects of biochar on soil properties and vegetable qualities in our subsequent studies. Thank you very much for your great support!

Li, B.; Bi, Z.; Xiong, Z. Dynamic responses of nitrous oxide emission and nitrogen use efficiency to nitrogen and biochar amendment in an intensified vegetable field in southeastern China. GCB Bioenergy 2017, 9(2), 400-413.

Schmidt, H.P.; Kammann, C.; Hagemann, N.; Leifeld, J.; Bucheli, T.D.; Sánchez Monedero, M.A.; Cayuela, M.L. Biochar in agriculture - A systematic review of 26 global meta-analyses. GCB Bioenergy 2021, 13, 1708-1730.

Zhang, C.; Zeng, G.; Huang, D.; Lai, C.; Chen, M.; Cheng, M.; Tang, W.; Tang, L.; Dong, H.; Huang, B.; Tan, X.; Wang, R. Biochar for environmental management: Mitigating greenhouse gas emissions, contaminant treatment, and potential negative impacts. Chem. Eng. J. 2019, 373, 902-922.

Zhang, Y.; Wang, J.; Feng, Y. The effects of biochar addition on soil physicochemical properties: A review. CATENA 2021, 202, 105284.

Reviewer 2 Report

This work deals with N2O and NO emissions, which were reduced by the addition of biochar.

Comments:

Introduction: It can be useful. It will improve the introduction. https://www.sciencedirect.com/science/article/pii/S0048969722011354

Good statement at the end of the introduction. Yes, it could be beneficial, but have you also considered the possible risks?

2. Materials and methods 

,,This experimental site, established in November 2016,, You are reporting 2016, why have you presented these results only now?

,,Henan Sanli New Energy Co., Ltd.,, Specify city and country.

,,2.2. Field design Before transplanting, urea (46.0% N), calcium magnesium phosphate (14.0% P2O5), and potassium chloride (63.2% K2O),, List the individual formulas of the preparations.

Line numbers should be added for better orientation.

Overall, the results section is well done.

Finally, add more opportunities and continuation of such research, as well as possible risks.

Table 1, 2 and 3: It is good when standard deviations are also given.

Author Response

(1) Introduction: It can be useful. It will improve the introduction.

https://www.sciencedirect.com/science/article/pii/S00489697 22011354

Good statement at the end of the introduction. Yes, it could be beneficial, but have you also considered the possible risks?

A: Thanks for your insightful comments. We read carefully the recent publication as recommended in Science of the Total Environment 826 (2022) 154043 entitled as “Herbal plants- and rice straw-derived biochars reduced metal mobilization in fishpond sediments and improved their potential as fertilizers” by Mehmood et al. We do learn from it that biochar has multiple benefits including stabilizing toxic elements in the sediments, which can be reutilized as fertilizers for crop production.

Besides those benefits, we did consider the possible risks of biochar amendments on Page 5 lines 64-70. The possible risk of biochar is mainly derived from the feedstock; therefore, feedstock with low concentrations of harmful substances should be selected for application into the soil, including wheat straw (Quilliam et al., 2013). Previous studies have shown that slow pyrolysis and low temperatures are recommended because of the lower harmful substances and ecological risks of biochar produced in this way (Xiang et al., 2021). The biochar used in our study was obtained by pyrolysis of wheat straw at a temperature range of 400-500°C for 6 hours, which meets the mentioned standard of low harmful substance. Although there might be possible risks for biochar field application, it will not obscure the positive impacts of biochar on soil properties, plant growth and microbial activity (Schmidt et al., 2021).

Mehmood,S., Ahmed,W., Ahmed J.M., et al. 2022. Herbal plants- and rice straw-derived biochars reduced metal mobilization in fishpond sediments and improved their potential as fertilizers. Science of the Total Environment 826, 154043.

Quilliam, R.S., Rangecroft, S., Emmett, B.A., Deluca, T.H., Jones, D.L., 2013. Is biochar a source or sink for polycyclic aromatic hydrocarbon (PAH) compounds in agricultural soils? Global Change Biology Bioenergy 5(2), 96-103.

Schmidt, H.P.; Kammann, C.; Hagemann, N.; Leifeld, J.; Bucheli, T.D.; Sánchez Monedero, M.A.; Cayuela, M.L. Biochar in agriculture - A systematic review of 26 global meta-analyses. GCB Bioenergy 2021, 13, 1708-1730.

Xiang, L., Liu, S., Ye, S., Yang, H., Song, B., Qin, F., Shen, M., Tan, C., Zeng, G., Tan, X., 2021. Potential hazards of biochar: The negative environmental impacts of biochar applications. Journal of Hazardous Materials 420, 126611.

 (2) 2. Materials and methods

This experimental site, established in November 2016, You are reporting 2016, why have you presented these results only now?

A: Yes, you are right! Our field experiments were established in November 2016 and we have been conducting the observations for soil, plant, and N2O and NO emissions regularly since the establishment. We would like to investigate the comprehensive effects of biochar amendments for several years. Given the current publications are mainly from short-term field studies (< 2 years) or incubations (Schmidt et al., 2021; Zhang et al., 2019; Zhang et al., 2021), our results are providing evidences for the medium-to-long term results of biochar amendments in greenhouse vegetable field.

Schmidt, H.P.; Kammann, C.; Hagemann, N.; Leifeld, J.; Bucheli, T.D.; Sánchez Monedero, M.A.; Cayuela, M.L. Biochar in agriculture - A systematic review of 26 global meta-analyses. GCB Bioenergy 2021, 13, 1708-1730.

Zhang, C.; Zeng, G.; Huang, D.; Lai, C.; Chen, M.; Cheng, M.; Tang, W.; Tang, L.; Dong, H.; Huang, B.; Tan, X.; Wang, R. Biochar for environmental management: Mitigating greenhouse gas emissions, contaminant treatment, and potential negative impacts. Chem. Eng. J. 2019, 373, 902-922.

Zhang, Y.; Wang, J.; Feng, Y. The effects of biochar addition on soil physicochemical properties: A review. CATENA 2021, 202, 105284.

(3) Henan Sanli New Energy Co., Ltd., Specify city and country.

A: Sorry for the inconvenience! We added the city and country on Page 8 lines 101.

(4) 2.2. Field design Before transplanting, urea (46.0% N), calcium magnesium phosphate (14.0% P2O5), and potassium chloride (63.2% K2O), List the individual formulas of the preparations

A: Sorry for the inconvenience! We added the individual formulas of the preparations in the table below (Table R1) and Supplementary Information.

Table R1 Main compounds and ingredient content of the chemical fertilizers applied

Fertilizers

Main compounds

Main ingredient content

Urea

(NH2)2CO

46.0% N

Calcium magnesium phosphate

Ca3(PO4)2

CaSiO3

MgSiO3

14.0% P2O5

45.0% CaO

20.0% SiO2

12.0% MgO

Potassium chloride

KCl

63.2% K2O

(5) Line numbers should be added for better orientation.

A: Sorry for the inconvenience! We added the line numbers.

(6) Overall, the results section is well done.

A: Thank you very much for your nice support.

(7) Finally, add more opportunities and continuation of such research, as well as possible risks.

A: Thanks for your suggestion. We added the corresponding expressions to nourish our conclusions on page 26 line 479-482.

(8) Table 1, 2 and 3: It is good when standard deviations are also given.

A: Sorry for the inconvenience! The data listed in Tables 1, 2 and 3 do not represent the means, but rather the F-values from the statistical analysis (Table 1), the correlations (Table 2) and the results of the principal component analysis (Table 3). The means and standard deviations for the specific three-year soil physicochemical properties are presented in Table S4. Thanks for your understanding.

Thank you all very much once again for your helpful comments and great support!

Round 2

Reviewer 2 Report

The manuscript can be accept.